# Data-Driven EEG Band Discovery with Decision Trees

**DOI:** 10.3390/s22083048

**Published:** 2022-04-15

**Authors:** Shawhin Talebi, John Waczak, Bharana A. Fernando, Arjun Sridhar, David J. Lary

**Affiliations:** Hanson Center for Space Sciences, University of Texas at Dallas, Richardson, TX 75080, USA; john.waczak@utdallas.edu (J.W.); ashen.fernando@utdallas.edu (B.A.F.); arjun.sridhar@utdallas.edu (A.S.); david.lary@utdallas.edu (D.J.L.)

**Keywords:** electroencephalography (EEG), EEG bands, decision tree, machine learning

## Abstract

Electroencephalography (EEG) is a brain imaging technique in which electrodes are placed on the scalp. EEG signals are commonly decomposed into frequency bands called delta, theta, alpha, and beta. While these bands have been shown to be useful for characterizing various brain states, their utility as a one-size-fits-all analysis tool remains unclear. The goal of this work is to outline an objective strategy for discovering optimal EEG bands based on signal power spectra. A two-step data-driven methodology is presented for objectively determining the best EEG bands for a given dataset. First, a decision tree is used to estimate the optimal frequency band boundaries for reproducing the signal’s power spectrum for a predetermined number of bands. The optimal number of bands is then determined using an Akaike Information Criterion (AIC)-inspired quality score that balances goodness-of-fit with a small band count. This data-driven approach led to better characterization of the underlying power spectrum by identifying bands that outperformed the more commonly used band boundaries by a factor of two. Additionally, key spectral components were isolated in dedicated frequency bands. The proposed method provides a fully automated and flexible approach to capturing key signal components and possibly discovering new indices of brain activity.

## 1. Introduction

The electrical activity produced by the brain was discovered by Richard Caton. Hans Berger later demonstrated that this activity could be recorded directly from the scalp [1]. This technique for measuring brain activity is called electroencephalography (EEG). It consists of an array of electrodes placed on the scalp that record fluctuations in electric potential arising from the activity of synchronized neural populations [2,3].

A popular method of analyzing EEG is spectral analysis. This consists of decomposing signals onto a frequency basis (Figure 1) and grouping frequencies into spectral bands (i.e., frequency ranges). A popular method for EEG signal decomposition is Welch’s method which estimates a signal’s spectral power density across a range of frequencies [4]. Commonly used spectral bands are: delta, theta, alpha, and beta [5].

EEG bands correspond to brain phenomena in specific brain areas and contexts. For example, alpha activity from occipital regions (i.e., visual cortex) in relaxed, awake animals track with eye closures [6]. During sleep, alpha-band activity is observed at sleep onset, also called sleep spindles (7–14 Hz), and delta waves (1–4 Hz) appear in deep sleep stages [6]. Additionally, EEG bands have been used in a variety of contexts such as: measuring cognitive load [7,8,9], disease diagnosis [10,11,12], and predicting emotions [13,14,15].

Despite the widespread use of established spectral bands (e.g., delta, theta, alpha, and beta), there are two potential concerns with the current standard. First, there is significant variability in band boundaries across studies, as shown in Figure 2. This disagreement may be a result of a variety of factors such as hardware, filtering, and experimental task [12]. Second, ideal band definitions may depend on individual characteristics such as age, genetics, personality, and task performance [16].

These concerns motivate the use of data-driven approaches for the discovery of optimal EEG band boundaries. Such an approach tailors EEG bands to a specific experimental context in an automated way. Many methodologies have been proposed to achieve this goal [16,17,18,19,20]. These approaches typically make use of a target variable to ground the optimization of band boundaries [17,18,19,20]. For example, learning the best choice of boundaries for classifying Alzheimer’s disease [17]. A more recent approach proposed by Cohen makes use of a generalized eigendecomposition of the covariance matrix for multi-channel EEG data [16].

Here, we present a new method of EEG band discovery, that makes use of decision trees, a popular machine learning framework. Optimal bands are inferred for an input EEG power spectrum in a self-supervised way. Two key points distinguish this method from past approaches.
Band discovery is completely self-supervised in the sense that only EEG data is usedAs the method only uses a power spectrum, it is agnostic as to how the data is generated, so it can handle both single- and multi-channel data in a variety of contexts.

## 2. Methods

### 2.1. Method Overview

An overview of the data-driven method for EEG band discovery is illustrated in Figure 3. The first step is to obtain single- or multi-channel EEG recordings. Second, a power spectrum is computed using, for example, Welch’s method [4]. It is noted that information from multiple EEG channels can be aggregated into a single power spectrum in this step. Third, a set of band boundaries is derived for every possible choice of band count. Possible values range from 2 bands up to the total number of unique frequency values in the power spectrum. Fourth, an AIC-inspired quality score is computed for each choice of band count. Finally, the band boundaries with the smallest quality score are selected as the best choices.

There are two key components of this data-driven method. The first component is the use of a decision tree to obtain optimal EEG band boundaries for a specified band count. There are two main benefits to using a decision tree in this context. First, due to the structure of decision tree regression, frequency values are grouped into true bins. In other words, frequency values in a discovered band are adjacent, which may not be guaranteed by other regression techniques. The second is the ease of use. There are many efficient and ready-to-use implementations of decision tree optimization across many computational frameworks [21,22,23,24].

The second key component is an Akaike Information Criterion (AIC)-inspired quality score which serves as an objective from which the choice of band count can be optimized. As a result, this objective eliminates the need for manual entry of a band count.

### 2.2. Decision Trees

Decision trees are a widely-used and intuitive machine learning approach. Typically, they are used to solve prediction problems. That is, identifying a discrete target class (classification) or estimating a continuous target value (regression) from a set of predictor variables [25].

Data can be used to *grow* decision trees in an optimization process called training. Training requires a training dataset, which consists of predictor variables labeled with target values. A standard strategy for training a decision tree is recursively partitioning data via a greedy search method. The search determines the gain from each splitting option and then chooses the one that provides the greatest gain [25,26]. Splitting options are the observed predictor variable values in the training dataset. Gain is determined by the split criterion e.g., Gini impurity or mean squared error (*MSE*).

For example, in a regression task, data records are recursively split into two groups such that the weighted average *MSE* of the target value is minimized from the resulting groups. *MSE* is defined as follows.
$$MSE=\frac{1}{N}\sum_{i=1}^{N}{\left(Y_{i}-{\hat{Y}}_{i}\right)}^{2}$$
where, *N* is the total number of observations in a given partition. $Y_{i}$ is the true target value for the *i*th frequency value. ${\hat{Y}}_{i}$ is the tree estimated target value for the *i*th frequency value. This splitting procedure can continue until all data partitions are pure, meaning every data record in a given partition corresponds to a single target value. Although this implies decision trees can be perfect estimators, such an approach would result in overfitting. Therefore, the trained decision tree would not perform well on data sufficiently different than the training dataset.

One way to combat the overfitting problem is hyperparameter tuning. Hyperparameters are values that constrain the growth of a decision tree. Common decision tree hyperparameters are the maximum number of splits, minimum leaf size, and the number of splitting variables. The key result of setting decision tree hyperparameters is to limit the tree’s size, which can help avoid predictions only suitable to the training dataset. In this work, we use decision tree hyperparameters to control the number of discovered frequency bands.

### 2.3. Band Discovery with Decision Trees

Optimal EEG frequency bands can be estimated using the decision tree framework. Here, *optimal* means the frequency groupings that best reproduce an input signal’s log spectral density for a set number of bands. To achieve this goal, a decision tree is used to solve a regression problem in the usual way. A visual overview of the decision tree training in this context is shown in Figure 4.

We use a single predictor variable (frequency) to estimate a single target variable (natural logarithm of the power spectral density). Any calculation of the power spectral density can be used and plugged into our technique. One such method is described by Welch [4]. Using this technique, for example, the target variable is defined by the following expression.
$$Y_{i}=\ln\hat{P}\left(f_{i}\right)$$
where, $Y_{i}$ is the *i*th element of the target variable array, $\hat{P}$ represents the spectral estimate according to [4], and $f_{i}$ is the *i*th element of the predictor variable array i.e., *i*th frequency value from the EEG signal decomposition.

The decision tree splits frequency values into subgroups and assigns each subgroup a single target value estimation. A greedy search of the decision tree parameter space yields frequency splits that best reproduce target values [25,26]. Thus, through this optimization process, we automatically obtain the optimal member-adjacent frequency bands for a predefined band count.

The band count corresponds to the maximum number of splits used in decision tree training. However, through the use of an AIC-inspired quality score, the proposed method removes the need for manual entry of this quantity. This is discussed further in the next section.

### 2.4. Quality Score for Band Boundaries

Although decision tree optimization can be leveraged to identify optimal EEG frequency bands, this method requires the number of bands to be predetermined. Instead of choosing a band count manually, here we describe an objective data-driven strategy. The choice of band count is framed as an optimization problem, where we define an objective that can be optimized with respect to the band count.

One choice of objective is the $r^{2}$ regression score. In this context, the $r^{2}$ value corresponds to how well a set of decision tree-derived EEG band boundaries reproduce an underlying power spectrum. While the decision tree optimization strategy described previously will ensure band boundaries are optimal for a given number of bands, different choices of band count will correspond to different $r^{2}$ values. An example of this is shown in Figure 5, where the $r^{2}$ regression scores of several different choices of band count are plotted for the same dataset.

However, the $r^{2}$ score is a problematic objective choice, since it strictly increases with the number of bands. Therefore, the maximum regression score would correspond to the largest possible number of bands i.e., a frequency “band” for every observed frequency value. One simple solution is to introduce an objective that incorporates both the $r^{2}$ regression score and a penalty for the number of bands. This is the goal of popular measures such as the Bayesian Information Criterion (BIC) and Akaike Information Criterion (AIC) [27]. Taking inspiration from AIC, we construct an empirically derived quality score (QS) to help choose a model that balances the best regression score while limiting the number of bands.

AIC is a measure of model quality, where smaller values imply better models [27,28]. It is defined in terms of the maximum value of the likelihood function for the model, *L*, and the number of parameters in the model, *k*.
$$AIC=-\log L^{2}+2k$$

The quality score (QS) we employed closely resembles AIC with two modifications. First, in lieu of the squared maximum likelihood value, we used the $r^{2}$ regression score. Since $r^{2}$ values are between [0,1], the first term in the QS equation below will be between [0,∞), however, this range is not very large in practice e.g., for *r*${}^{2}\geq 0.135$, the first term is approximately between [0,2]. Second, we divided the second term by *N*, where *N* is the maximum number of bands, or equivalently, the total number of observed frequency values. This ensures the second term in the equation below takes values in the range [0,2].
$$QS=-\log r^{2}+2k/N$$

QS provides a way to compare EEG band boundaries in a way that accounts for both goodness-of-fit and band count. It will typically take values between 0 and 2, where smaller values correspond to better models. By computing the QS for every possible band count, we can choose the best EEG band boundaries as the choice with the smallest QS.

Although QS takes inspiration from AIC, a theoretically grounded quantity, its derivation is empirical, therefore it may not be most suitable for all applications. Furthermore, there are countless other objective choices to optimize band count. The decision tree method described in Section 2.3 is independent of this band count optimization step, and thus can be enhanced by a variety of choices.

### 2.5. Software Implementation

This two-part technique is implemented using the Sci-Kit learn Python library, a popular and free machine learning software [21]. The decision tree implementation used is the sklearn.tree.DecisionTreeRegressor class. The chosen parameters for the decision tree training are specified in Table 1. A detailed description of each parameter can be found at the Sci-Kit learn documentation: https://scikit-learn.org/stable/modules/generated/sklearn.tree.DecisionTreeRegressor.html (accessed on 6 March 2022).

Our code is open-source and publicly available at the GitHub repository: https://github.com/mi3nts/decisiontreeBinning (accessed on 6 March 2022). Although Python is used for our implementation, other statistical software packages can be readily used to implement this method [22,23,24].

## 3. Results

In the following subsections, we explore two case studies that apply the proposed data-driven method for EEG frequency band discovery to an artificial and experimental dataset, respectively. A Python script to reproduce both case studies is freely available at the following GitHub repository: https://github.com/mi3nts/decisiontreeBinning (accessed on 6 March 2022).

### 3.1. Case Study 1: Artificial Data

As a first demonstration of the method, we produce an artificial EEG power spectrum as shown in Figure 6. The spectrum consists of the characteristic 1/f shape for EEG signals with added white noise. A mathematical expression for the artificial power values is given below.
P=1/f+r
where, *P* is the artificially generated power value. *f* is the frequency value. *r* is a uniformly distributed random value between 0 and 0.4.

The results of applying decision tree-based band discovery to the artificial power spectrum are shown in Figure 7 for 5 different choices of band count. In the top 5 plots, the true power spectrum is shown as a solid blue line, the decision tree estimated spectrum is plotted as a dashed orange line, and the discovered band boundaries are indicated by dashed vertical red lines. The plots are titled according to the number of bands and $r^{2}$ (coefficient of determination) regression score. The $r^{2}$ score indicates how well the discovered bands reproduce the true power spectrum. As a comparison, the typical boundaries of the delta, theta, alpha, and beta bands according to the review by [12] are shown at the bottom of Figure 7. Each band is labeled with text. The $r^{2}$ score of the standard bands is computed by comparing the average power value within each band with the true values. This value is provided in the plot title.

The greedy search algorithm used in decision tree regression preserves band boundaries when new bands are added. In Figure 7, for example, 7.3 Hz is a band edge in every case (i.e., from 2 bands to 6 bands). It is interesting to note that the discovered 4 bands case is nearly identical to the typical delta, theta, alpha, and beta band boundaries according to [12]. Thus, it may be that the typical band boundaries are a good representation of this characteristic power spectrum.

In Figure 7, as more bands are added, the $r^{2}$ regression score increases. A diagrammatic representation of this observation is shown in Figure 5, where model regression scores are plotted against the number of bands. Since there are 150 unique frequency values in this first artificial dataset, the maximum number of bands is 150. Colored dashed vertical lines indicate band choices that exhibit a large jump in the $r^{2}$ score.

Since the $r^{2}$ score strictly increases with the number of bands, using it as an objective from which to choose the band count would always result in a “band" for every observed frequency value. However, the AIC-inspired quality score (QS) defined in Section 2.4 does not suffer from this issue. This is illustrated in Figure 8, which plots QS against the number of bands. Additionally, the *r*${}^{2}$-based fitness term in QS is shown as a dashed blue line, the band count penalty term is plotted as a dashed orange line, and the minimum QS value is indicated by a yellow star. A minimum QS value is observed at 6 bands, implying the best choice of band count for this spectrum is 6.

The top plot in Figure 9 outlines the optimal bands based on a quality score (QS) minimization strategy. The plot title indicates the number of bands (6), $r^{2}$ regression score (0.94), and the quality score of the band definitions (0.14). The bottom plot in Figure 9 similarly outlines the standard bands, titled with the same metrics. The QS of the standard bands is computed using the QS equation in Section 2.4 with *k* = 4. Although the discovered bands include more parameters, the QS is about half of that of the standard bands, thus it is a better characterization of the underlying spectrum based on this objective. Based on the data-driven approach, new band boundaries are discovered that complement the standard delta, theta, alpha, and beta bands by dividing the standard delta and beta bands into two.

### 3.2. Case Study 2: Experimental Data

We evaluate the band discovery method on experimental data from the PhysioNet dataset: EEG During Mental Arithmetic Tasks [29,30]. EEG data were collected monopolarly using the Neurocom EEG 23-channel system (Ukraine, XAI-MEDICA). The electrodes were placed on the scalp according to the International 10/20 montage. Interconnected ear electrodes were used as the reference. A 30 Hz cut-off frequency high-pass filter and a 50 Hz power line notch filter were used. The data are artifact-free segments of 60 s. In preprocessing, Independent Component Analysis (ICA) was used to eliminate artifacts (eyes, muscles, and cardiac). For this case study, the baseline EEG recording from Subject 00 is used. Occipital electrodes (O1 and O2) are averaged to produce an aggregated occipital EEG signal.

The aggregated occipital time-series signal and its corresponding power spectrum are shown in Figure 10. Welch’s method using a Hanning window and a segment length of 1028 estimated the power spectral density of the aggregated signal [4]. Due to the signal preprocessing scheme used here, the power spectrum does not follow the typical 1/f shape. Nevertheless, an alpha rhythm peak is observed. Although this experimental power spectrum is characteristically different than the previous artificial spectrum, the EEG bands discovered by our data-driven approach will automatically adapt to it.

We repeat the two-part strategy from Case Study 1. First, we derive band boundaries using the decision tree strategy for every possible choice of band count (i.e., 2 to 60 bands). Second, we use the quality score (QS) to identify the best number of bands. The QS is plotted against the number of bands in Figure 11. The minimum QS value occurs for the 6 bands case.

Figure 12 compares the bands discovered by applying the proposed band discovery strategy to the experimental data (top plot), the optimal bands discovered in Case Study 1 (middle plot), and the standard EEG band boundaries from [12] (bottom plot). The discovered bands from the experimental data (top plot) outperforms the other band choices, with both a significantly higher $r^{2}$ score and lower (better) QS. For the present case study, the quality score for the discovered bands was 0.35, compared to scores of 0.86 and 0.8 for the bands discovered in Case Study 1 and standard EEG bands, respectively. Despite the fact that the discovered band boundaries outperformed the standard bands in Case Study 1, when the same boundaries are applied to new experimental data, the standard bands perform better. The poor performance of the bands from Case Study 1 in characterizing this experimental power spectrum highlights the need to tailor EEG bands to specific datasets.

The bands identified from the experimental data isolate spectral features. Specifically, the peak in power spectral density between 10 and 12 Hz is partitioned into a dedicated band. This gives an idea of how the proposed method works. It will tend to learn frequency bands that correspond to peaks in the underlying power spectrum.

## 4. Discussion

This paper outlines a self-supervised method for discovering optimal EEG frequency bands. It differs from previous methods in two important ways. First, band discovery is entirely self-supervised, so an external target variable is not necessary. Second, since the method solely uses a power spectrum, it is capable of handling both single- and multi-channel data across contexts.

The methodology was evaluated by using two case studies. In the first case study, the method was applied to a power spectrum consisting of a 1/f shape with white noise added. The discovered bands overlapped with the typical delta, theta, alpha, and beta boundaries. Two additional bands were found within the typical delta and beta frequencies. Despite the larger number of parameters, the discovered bands had a quality score that was nearly half that of the typical ones, thus indicating significantly better performance. In the second case study, the method was applied to a baseline EEG recording from the open-access PhysioNet dataset: EEG During Mental Arithmetic Tasks [29,30]. As in the previous case, the discovered EEG bands significantly outperformed the more conventional boundaries. Additionally, the discovered bands isolated a peak in the power spectral density curve into a dedicated frequency band.

The proposed method has two key strengths. First, the method provides a way to determine frequency bands that are representative of an underlying power spectrum while keeping the number of bands to a minimum. This results in a parameter-free and reproducible approach to the discovery of optimal EEG bands. Second, the method is readily accessible since it is based on decision tree optimization, which has many efficient and ready-to-use implementations [21,22,23,24]. Additionally, we made our implementation of the technique open-source and publicly available (https://github.com/mi3nts/decisiontreeBinning (accessed on 6 March 2022)).

Unlike other methods that optimize band boundaries to estimate a particular variable (e.g., disease diagnosis), our approach relies only on power spectral density values for estimation. Although this can be considered a strength, it also has a downside. Since EEG bands are purely based on spectral density curves, their interpretation may not be clear. Consequently, interpretation of bands discovered using this method may require additional effort compared to other approaches.

In Table 2, we present a comparison of different approaches to EEG band discovery. Five previous methods are compared with the one proposed in this article [16,17,18,19,20]. The comparisons are based on four characteristics: supervised, self-supervised, single-channel, and multi-channel. A supervised method is one that uses a target variable to ground the band optimization, such as disease diagnosis, stimulus details, and task type. Conversely, a self-supervised method uses the inherent characteristics of EEG signals to partition bands. Single-channel means the technique can operate on single-channel EEG data, while multi-channel indicates the technique operates on multi-channel EEG data. The uniqueness of our approach resides in the fact that it is a self-supervised approach that works on both single-channel and multi-channel data. Furthermore, it is the first EEG band discovery method that uses the decision tree machine learning framework.

A key feature of the presented approach is that it is agnostic to *how* the input power spectrum is generated, thus it can readily be applied to other types of power spectra (e.g., audio signals, hyperspectral imaging). Hyperspectral imaging, for instance, captures images with layers beyond the standard red, green, and blue. This provides a power spectrum for each pixel of a hyperspectral image. Using the proposed band discovery method, interesting spectral features in hyperspectral images can be detected in a self-supervised way.

Additionally, this method can be applied to other types of input variables. For example, a times series (e.g., heart rate over time) could be used in lieu of a power spectrum. This would result in the discovery of temporal epochs, as opposed to frequency bands.

## 5. Conclusions

EEG serves as a window to underlying neural processes. Spectral analysis of EEG examines the oscillations in electric potentials arising from the brain. Despite the widespread use of established delta, theta, alpha, and beta bands for EEG, their boundaries vary widely across studies, which may be a result of variations in experimental details and participant differences. This motivates the use of objective and data-driven approaches to EEG band discovery.

In this work, we leveraged the readily available optimization of a decision tree for regression to discover EEG bands most appropriate for a given dataset and a predetermined number of bands. The best choice of band count was then determined using an AIC-inspired quality score. We applied the presented method to both artificial and open-access experimental data. Discovered bands isolated spectral features into dedicated bands and outperformed the standard band definitions. Data-driven EEG band discovery may provide new indices of neural activity which can adapt to a variety of experimental and subject characteristics.

## Figures and Tables

**Figure 1 sensors-22-03048-f001:**
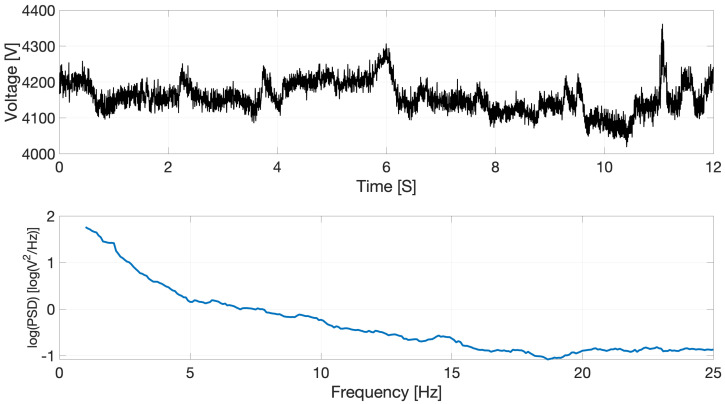
(**Top**) Example EEG time series signal sampled at 500 Hz. (**Bottom**) EEG signal’s corresponding power spectrum, where the natural logarithm of the signal’s power spectral density (PSD) is plotted against frequency.

**Figure 2 sensors-22-03048-f002:**
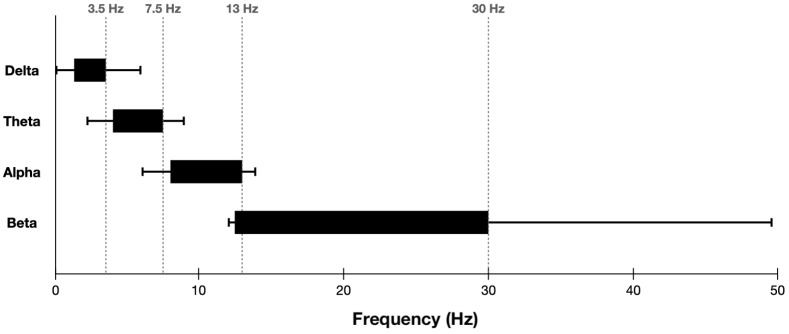
Box plot illustrating variability between delta, theta, alpha, and beta band boundaries across studies. Boxes indicate the typical frequency range of each band. Whiskers represent the smallest and largest band edges observed across studies. Plot adapted from figure in [12].

**Figure 3 sensors-22-03048-f003:**
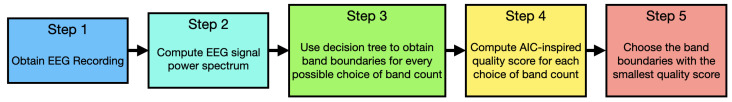
Conceptual overview of the data-driven methodology used in this paper.

**Figure 4 sensors-22-03048-f004:**
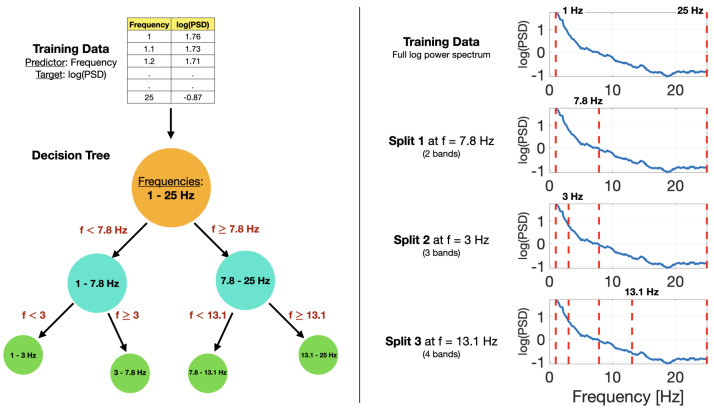
(**Left**) Visual summary of a decision tree partitioning frequency values based on the natural logarithm of the power spectral density. (**Right**) Visualization of decision tree splits of frequency values with power spectra.

**Figure 5 sensors-22-03048-f005:**
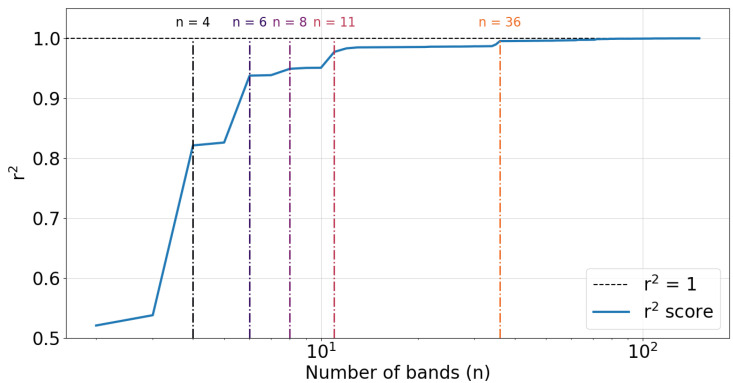
$r^{2}$ regression scores plotted against the number of frequency bands included in the decision tree model. The data used to derive these bands and $r^{2}$ values is the artificial data described in Case Study 1 in Section 3.1. Colored dashed vertical lines highlight large jumps in $r^{2}$ and are labeled by the corresponding number of bands.

**Figure 6 sensors-22-03048-f006:**
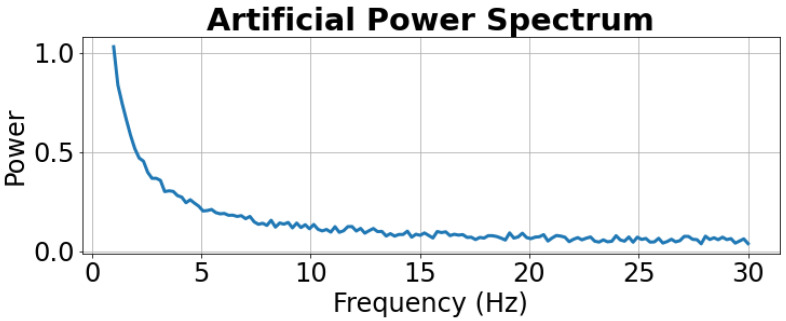
Artificial power spectrum for initial demonstration of data-driven method. The spectrum consists of the characteristic 1/f curve for EEG signals with added white noise.

**Figure 7 sensors-22-03048-f007:**
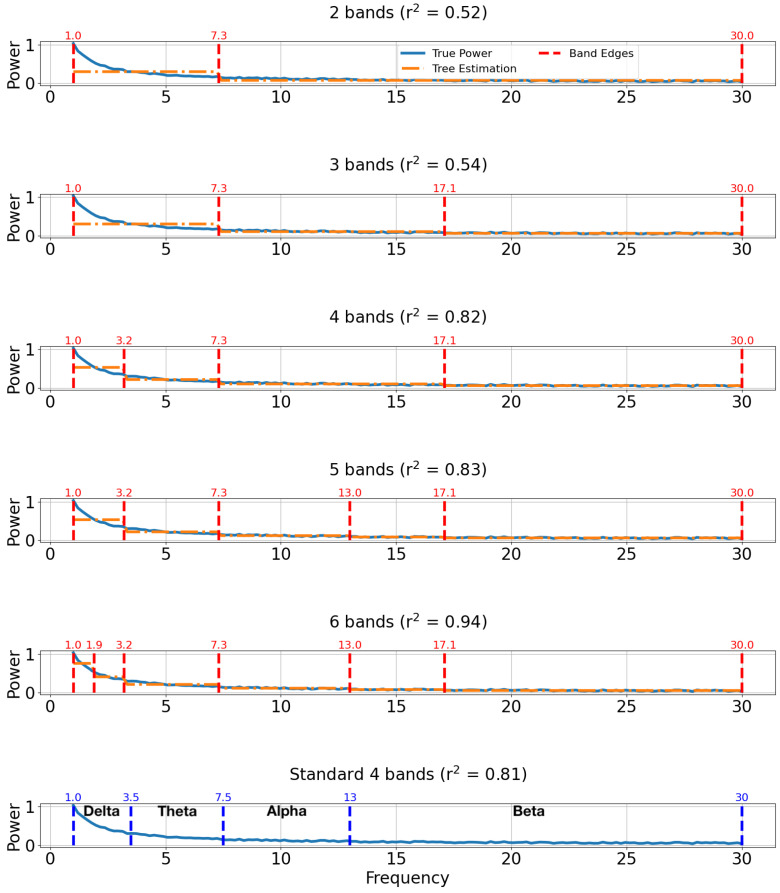
Band comparisons for artificial power spectrum. The true power spectra are plotted with solid blue lines, predicted spectra are plotted with dashed orange lines, and discovered band boundaries are indicated by dashed vertical red lines. The plots are titled according to their number of bands and $r^{2}$ regression score. For comparison, typical values of the standard 4 bands (delta, theta, alpha, and beta) according to [12] are shown in the bottom plot along with the true power spectrum plotted again as a solid blue line.

**Figure 8 sensors-22-03048-f008:**
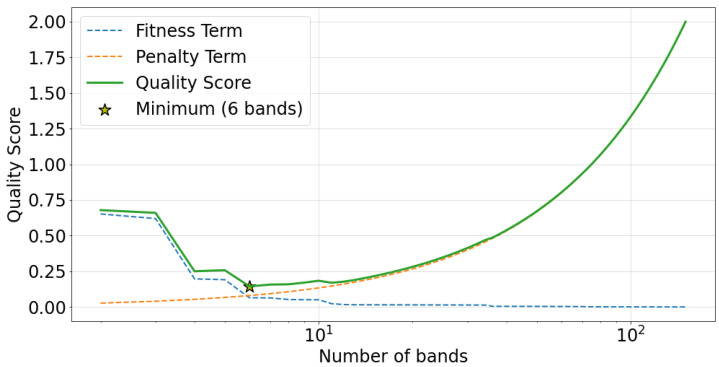
Empirically derived quality score (QS) plotted against the number of bands for a case study of an artificially generated power spectrum. The *r*^2^-based fitness term in QS is shown as a dashed blue line, the band count penalty term is plotted as a dashed orange line, QS is plotted as a green line, and the minimum QS value is indicated by a yellow star.

**Figure 9 sensors-22-03048-f009:**
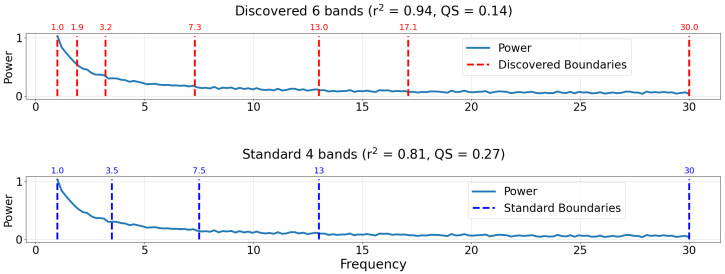
Comparison of discovered and standard bands for the case study of an artificially generated power spectrum. Plots are titled by the number of bands, $r^{2}$ regression score, and the quality score of the respective band boundaries. The true power spectrum is plotted as a solid blue line. (**Top**) Discovered bands using the proposed decision tree method employing a minimum quality score (QS) technique. Discovered band boundaries are indicated by dashed vertical red lines. (**Bottom**) Typical standard band boundaries are taken from review by Newsom [12]. Standard band boundaries are indicated by dashed vertical dark blue lines.

**Figure 10 sensors-22-03048-f010:**
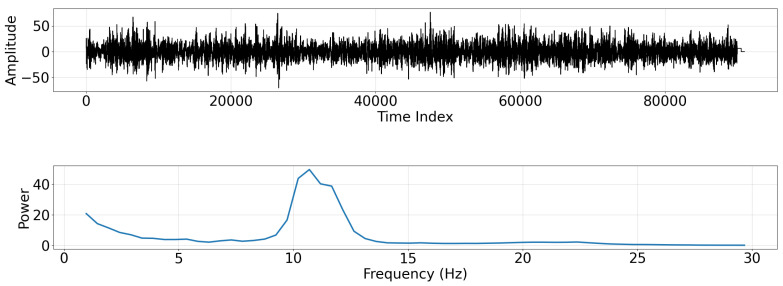
(**Top**) Time series of aggregated occipital EEG signal. (**Bottom**) Power spectral density plotted against frequency for aggregated occipital EEG signal plotted from approximately 1–30 Hz.

**Figure 11 sensors-22-03048-f011:**
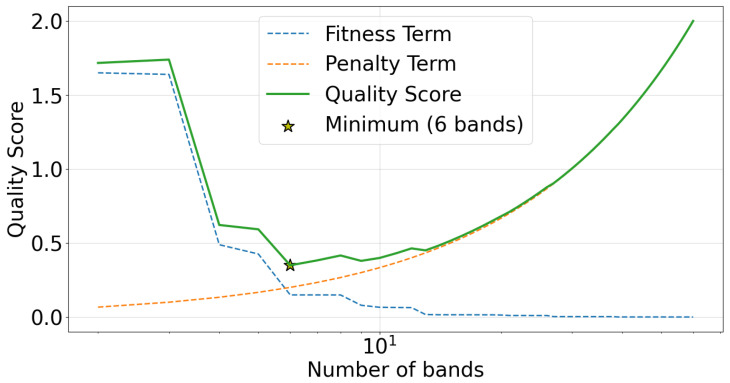
Empirically derived quality score (QS) plotted against the number of bands for the case study of experimental EEG data. The r^2^-based fitness term in QS is shown as a dashed blue line, the band count penalty term is plotted as a dashed orange line, QS is plotted as a green line, and the minimum QS value is indicated by a yellow star.

**Figure 12 sensors-22-03048-f012:**
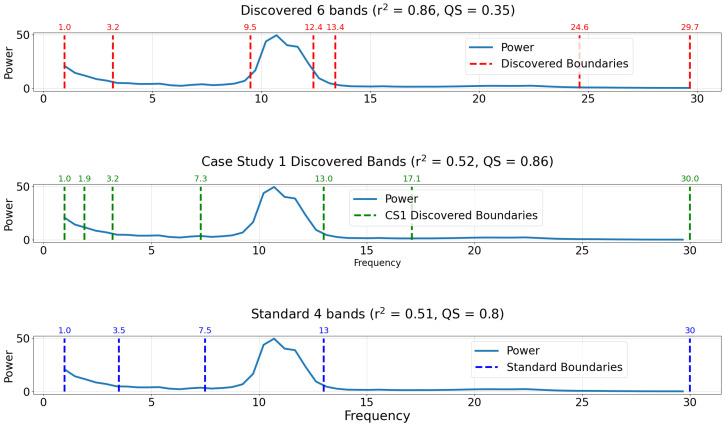
Comparison of discovered and standard bands for the case study of experimental EEG data. Plots are titled by the $r^{2}$ regression score and the quality score of the respective band boundaries. The true power spectrum is plotted as a solid blue line. (**Top**) Discovered bands using the proposed decision tree method employing a minimum quality score (QS) technique. Discovered band boundaries are indicated by dashed vertical red lines. (**Middle**) Discovered bands derived from artificial power spectrum in Case Study 1. Discovered band boundaries from Case Study 1 are indicated by dashed vertical green lines. (**Bottom**) Typical boundaries of standard bands are taken from a review by Newsom [12]. Standard band boundaries are indicated by dashed vertical dark blue lines.

**Table 1 sensors-22-03048-t001:** Table specifying the sklearn.tree.DecisionTreeRegressor parameters used in decision tree-based band discovery method. Parameter details can be found at the Sci-Kit learn documentation [21].

Name	Value
criterion	“squared_error”
splitter	“best”
max_depth	None
min_samples_split	2
min_samples_leaf	1
min_weight_fraction	0.0
max_features	None
random_state	None
max_leaf_nodes	Optimized with QS
min_impurity_decrease	0.0
ccp_alpha	0.0

**Table 2 sensors-22-03048-t002:** Tabular comparison of different data-driven approaches to EEG band discovery [16,17,18,19,20]. Methods are compared via four characteristics: supervised, self-supervised, single-channel, and multi-channel, shown as columns. Rows indicate the article reference outlining the approaches. The method proposed in this article is shown on the bottom row with the reference name “Proposed Method”. An “X” indicates the corresponding method has the listed characteristic.

References	Supervised	Self-Supervised	Single-Channel	Multi-Channel
Elgendi et al. (2011) [17]	**X**		**X**	**X**
Lee et al. (2012) [18]	**X**		**X**	**X**
Magri et al. (2012) [19]	**X**		**X**	**X**
Raza et al. (2015) [20]	**X**			**X**
Cohen (2021) [16]		**X**		**X**
Proposed Method		**X**	**X**	**X**

## Data Availability

Artificial dataset used in this work can be generated using software provided at the GitHub repository: https://github.com/mi3nts/decisiontreeBinning (accessed on 6 March 2022). Experimental data is from the PhysioNet dataset: EEG During Mental Arithmetic Tasks https://physionet.org/content/eegmat/1.0.0/ (accessed on 6 March 2022) [29,30].

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
