# Peer review of "Data-Driven EEG Band Discovery with Decision Trees"

_sensors, 2022, doi:10.3390/s22083048_

Round 1

Reviewer 1 Report

The authors should add more comments about figures 6, 8, 11.
In the introduction part, the authors should add some comments about already used methods for decomposition of the signals.
What sampling frequency was used for a signal from figure 1?
What represents the acronym PSD in figure 1, axis Oy?
The discussions part is very briefly presented.
The authors should add a comparison with other approaches (discovery EEG band using other algorithms/methods) from the scientific literature.

Author Response

Please find responses in attached PDF.

Reviewer 2 Report

This study aimed to propose a data-driven methodology for determining the best EEG bands for a given dataset. I have the following major suggestions.

  1. The abstract should be rewritten and improved by combining the objectives, short methodology, main findings, numerical form of results, and prospective application
  1. What is the novelty of this study although several approaches of optimal frequency band boundaries have been proposed earlier? Please write down the contribution of the study at the end part of the Introduction section in bulleted form.
  2. Section 2.1 is unnecessary. It need to be meged with introduction.
  3. Authors should describe the EEG numerous applications, such as mental workload, disease prediction, stress, emotion, brain-stimulation. EEG biomarkers are investigated for stroke prediction in article, healthsos: real-time health monitoring system for stroke prognostics.
  4. Which decision tree method authors used in this study?
  1. Authors should discuss the case studies of EEG applications. Task-Induced EEG-biomarkers are studied for post-stoke patients and healthy adults in article, quantitative evaluation of task-induced neurological outcome after stroke.
  2. Authors should give supporting mathematical expression, references in methodology to describe their approach.
  3. EEG was investigated in Brain Stimulation for different neurological workloads in article, quantifying physiological biomarkers of a microwave brain stimulation device.
  4. What is authors' motivation to partition the EEG bands in proposed segments? How this segmentation is beneficial rather than state-of-art EEG bands, alpha, beta, theta, delta, mu, and so on.
  5. EEG biomarkers was explored in different scenarios of driving states in a virtual driving environment in article, driving-induced neurological biomarkers in an advanced driver-assistance system.
  6. Authors should add a conceptual diagram to demonstrate their complete data-driven EEG band methodology.
  1. Authors should mention more details of EEG dataset used in this study, such as the ground and reference position, EEG data acquisition device.
  2. Which data compression or resampling methods authors proposed in their approach?
  1. The authors need to mention the model parameters or hyperparameters of their proposed method.
  1. A discussion section needs to be added. Authors should add references to the case studies mentioned above. Authors should discuss the strength and weaknesses of the proposed method with other studies in the discussion section.
  2. From the writing point of view, the manuscript needs to be checked for typos and the grammatical issues should be improved.

Author Response

Please find responses in attached PDF.

Round 2

Reviewer 1 Report

The authors should add a comparison table a the end of the discussion part with other approaches (discovery EEG band using other algorithms/methods and performance), from the scientific literature.

Reviewer 2 Report

Thanks for submitting the review, although most of the comments are not addressed. Here is my opinion in the authors' response regarding my comments in 1st review:

  1. Comments 1: Abstract doesn’t represent objectives, short methodology, main findings, the numerical form of results, and prospective application
  2. Comments 2: The novelty of this study is not clear yet. What is the necessity of a new EEG band boundary although state-of-art EEG bands already exist?
  3. Comment 7: Authors don’t report the supporting mathematical expression, such as EEG power spectral features methodology.
  4. Comment 9: Authors' motivation to partition the EEG bands in proposed segments doesn’t support EEG domain knowledge. The authors’ approach of EEG band discovery has no validation in prospective EEG analysis.
  5. Comment 14: Model parameters or hyperparameters of the proposed method were not appropriately reported.
  6. Comment 15: No comparative discussion was not reported in this manuscript. EEG domain knowledge was poorly written in this manuscript.

Round 3

Reviewer 2 Report

Authors should improve the figure quality in the manuscripts. 

Author Response

Please see attached file with comment responses.
